# Muscle Wasting among Hospitalized Children: A Narrative Review of the Feasibility and Accuracy of Diagnostic Methods

**DOI:** 10.3390/children10050795

**Published:** 2023-04-28

**Authors:** Sheikha AlQahtani, Dara Aldisi

**Affiliations:** 1Department of Community Health Sciences, Collage of Applied Medical Sciences, King Saud University, Riyadh 11451, Saudi Arabia; 2Department of Dietetics, Prince Sultan Military Medical City, Riyadh 11159, Saudi Arabia

**Keywords:** handgrip strength, feasibility, accuracy, muscle wasting, muscle mass, muscle strength, diagnostic methods, hospitalized children

## Abstract

Muscle wasting is associated with a worse quality of life and increased morbidity and mortality among hospitalized children, especially those with chronic diseases. This review was conducted to summarize the data available on the most feasible and accurate diagnostic methods for detecting muscle wasting among hospitalized children, especially to review the evidence of the accuracy and feasibility of conducting assessments using handgrip strength (HGS). Many diagnostic methods are used in hospital settings to assess muscle wasting, by evaluating either muscle mass or strength, with evidentiary support for assessing muscle mass provided using ultrasonography, magnetic resonance imaging, computed tomography, dual-energy X-ray, bioelectrical impedance analysis, and anthropometry measurements. Currently, the most common diagnostic method used to detect muscle strength loss is the handheld dynamometer. Studies support using HGS among healthy and hospitalized children to assess the overall nutritional status and especially muscle function. However, almost all of these studies have been conducted in hospital settings and recruited children with different chronic diseases using a small sample size. More longitudinal cohort studies with large sample sizes are needed to assess the accuracy and feasibility of using HGS among hospitalized children.

## 1. Introduction

Skeletal muscle (SM) is a foundation of human function [1]. Inadequate muscular skeletal muscle mass and strength in children are related to negative health outcomes. Children and adolescents with inadequate skeletal muscle mass and strength are more likely to experience metabolic dysfunction and cardiovascular illnesses, according to several studies [2,3,4,5,6]. Additionally, hospitalized children may suffer from muscle wasting, which can potentially impact how well they recover in the intensive care unit (ICU) and their long-term growth and development. To properly focus on optimum dietary and physical therapies to prevent muscle wasting and the likelihood of poor outcomes among these high-risk children, it is crucial to identify muscle wasting in hospitalized children [7,8].

Muscle-wasting disorder (MWD) is associated with a worse quality of life and increased morbidity and mortality [1]. The lack of a unified definition for this affliction is likely a cause underlying the underestimation of MWD prevalence among hospitalized children. One study reported the prevalence of MDW as 1.7%, which means that muscle wasting is not particularly prevalent; however, it still represents a significant feature of critical illness in children [8]. MWD is defined by the National Cancer Institute as the weakness, shrinkage, and loss of muscle caused by illness or inactivity [9]. Additionally, the Society of Sarcopenia, Cachexia, and Wasting Disorders (SCWD) defined MWD as any progressive acute or chronic medical condition characterized by weight loss that involves a significant loss of lean body mass, fat body mass, and muscle strength [1]. Sarcopenia and cachexia are two disorders characterized by decreases in skeletal muscle mass and strength, which are the most common causes of MWD [1]. The World Health Organization (WHO) defined muscle wasting as weight-for-age less than −2 standard divisions (z-score) from the median WHO growth standards or a mid-upper arm circumference (MUAC) of less than 115 mm [10,11]. Muscle strength is an indicator of muscle function and is associated with muscle mass, so this factor is considered a primary indicator of MWD [12,13,14]. Previous studies supported the association between muscle strength and mass, and the decline in muscle strength appears faster than the decline in muscle mass [15]. Muscle strength or power per unit of muscle mass is known as muscle quality [15]. MWD is classified based on etiology, disease severity, or the progression to acute or chronic MWD (Figure 1) [16].

Many diagnostic methods used to assess the overall nutritional status include muscle status. To overcome the clinical obstacles that limit the accuracy of estimating MWD among hospitalized children specific to the patient’s situation, researchers determine the optimal diagnostic methods to assess muscle mass, such as ultrasonography [17], magnetic resonance imaging (MRI) [18], computed tomography (CT) [19], dual-energy X-ray (DXA) [20], bioelectrical impedance analysis (BIA) [21], and anthropometry measurements [22]. The most common diagnostic tool used to detect muscle strength loss is a handheld dynamometer (HHD) [23]. 

In the present review, we provide an overview of the feasibility and precision of diagnostic methods for MWD, especially handgrip strength, to delay the progression of MWD as a primary method and detect weakness in muscle strength among hospitalized children. 

In this review, we followed a search strategy that identified all publications relevant to our research questions (RQs) below.

RQ1: What currently available evidence supports diagnostic methods, especially handgrip strength, in detecting muscle wasting among hospitalized children?RQ2: How feasible is it to use diagnostic methods, especially handgrip strength, to assess muscle wasting among hospitalized children?RQ3: What is the precision of diagnostic methods, especially handgrip strength, in detecting muscle wasting among hospitalized children?

## 2. Materials and Methods

We conducted a narrative review with a comprehensive exploration of diagnostic methods for detecting muscle wasting. A systematic literature search was performed in the following bibliographic databases: PubMed, Wiley Online Library, Google Scholar, and Web of Science. The searches were carried out in October 2022. Search terms were combined by Boolean logic (AND, OR). The keyword combinations used included the following keywords: (“hospitalized children” OR “pediatric*”) AND (“undernutrition*” OR “malnutrition*” OR “muscle wasting*”) AND (“diagnostic methods*” OR “*ultrasonography*” OR “magnetic resonance imaging*” OR “MRI*”) OR (“computed tomography*” OR “CT*” “dual-energy X-ray*” OR “DXA*” OR “bioelectrical impedance analysis*” OR “BIA*” OR “anthropometry measurements*” OR “handgrip strength*) AND (“feasibility*” OR “accuracy*”) AND (“muscle mass*” OR “muscle strength*”). The reference lists of the identified studies were manually searched for potentially relevant studies. Manually, duplicate records were removed. Studies were first screened by title and abstract and then by full text. Articles were required to be peer-reviewed, in full text, and in the English language. Inclusion and exclusion criteria were applied to the studies to ensure that their foci aligned with the purpose of this review. Those who were inappropriate for answering the research questions and/or did not fit the inclusion criteria were excluded. Studies eligible for inclusion were studies in English and studies conducted between January 2000 and September 2022; all study designs were included. From a total of 900 papers, there were 232 screened studies and 23 selected for final inclusion in our review. An overview of the search and screening process is provided in Figure 2.

## 3. Diagnostic Methods to Detect Muscle-Wasting Disorder (MWD)

Many MWD diagnostic methods exist, but the targeting of muscle function, based on mass or strength, is one of the most commonly used assessment criteria. Many diagnostic methods are used to detect muscle mass loss, such as ultrasonography, which is a noninvasive and real-time tool for visualizing normal and diseased muscle tissue, although the use of this tool is restricted for some patients with fluid–electrolyte imbalances [17,24,25]. In a prospective study comparing the health of full-term with that of preterm infants, an assessment of muscle wasting using ultrasonography showed lower skeletal muscle mass in the preterm group. Additionally, muscle thickness was measured with electronic calipers in the proximal and distal districts, which was statistically significantly thinner in prematurely born compared with term-born infants [26]. Another case–control study assessing muscle wasting via ultrasonography grayscale statistics, which are used to characterize images and determine the mean echo intensity (EI) and the size and number of spatially related homogenous areas, showed a significant difference in skeletal muscle composition between children with spastic cerebral palsy and their typically developing peers [27]. Muscle ultrasounds may be useful in detecting muscle wasting in hospitalized children, but they have not been shown to be sufficiently reliable for this population [28].

The second method is MRI, a noninvasive tool for identifying MWD, which uses a strong magnetic field, radiofrequency pulses, and a computer to provide precise images of the internal body structures, including muscle mass [29,30]. One advantage of an MRI is that radiation is not used. This tool is considered the gold standard but remains a costly method for providing an accurate measure of muscle mass [31,32]. A cohort study using MRI to estimate muscle mass in pediatric patients with liver disease by measuring the MRI-based total psoas muscle surface area (tPMSA) and correcting for height (tPMSA index = tPMSA/height2) found that tPMSA was independently associated with both the MRI results and the histological features of hepatic steatosis severity in children [33]. Another study used MRI with pediatric Duchenne muscular dystrophy (DMD) patients to assess the structural and functional diaphragm and associated pulmonary changes, concluding by recommending the use of MRI as a noninvasive tool for the functional and structural assessment of the diaphragm, which is a major muscle involved in respiration [18].

The third method is computed tomography, also known as a CT or CAT scan, which is a type of diagnostic imaging tool used for MWD [34]. The results of a CT scan provide three-dimensional, precise, accurate images of SM. However, this tool remains a costly method, and patients exposed to radiation may require sedation [35,36]. A retrospective examination of cross-sectional areas of skeletal muscle, intermuscular adipose tissue (IMAT), visceral adipose tissue (VAT), subcutaneous adipose tissue (SAT), and skeletal muscle density at the level of the third lumbar vertebra via CT scans obtained during the standard care of pediatric patients with high-risk neuroblastoma to assess the direction and timing of changes in skeletal muscle during treatment revealed small increases in skeletal muscle, skeletal muscle density, and IMAT. Moreover, a rapid increase in VAT and SAT was observed at an early point in treatment, with the highest volumes seen after six cycles of chemotherapy [19].

The fourth method is bone densitometry, also known as dual-energy X-ray absorptiometry or DXA, which produces internal images of the body, often of the hips and lower spine, to assess lean body mass [37]. DXA is a new and practical approach for estimating SM in pediatric subjects [38] and is considered a simple, quick, noninvasive procedure that does not require anesthesia, although it may require sedation, as the radiation dose for this procedure varies [37]. SM estimated via whole-body MRI has been used as a reference, and DXA has been utilized to measure total-body SM in children with different pubertal maturation stages, demonstrating that the DXA SM prediction model was valid for children at Tanner stage 5. However, significant overestimation occurred among children below Tanner stage 5. New SM prediction formulas have been developed using appendicular lean soft tissue estimates with DXA as the main predictor variable [20]. DXA measures of changes in lean mass were modestly associated with MRI measures of change in muscle volume, although there are several advantages to using DXA for the measurement of lean mass [39].

The fifth method is bioelectrical impedance analysis (BIA), which is extensively utilized by scientists and clinicians as a noninvasive and safe way of estimating body composition and body water content in children [21,40,41]. A study evaluating the level of agreement between body composition measurements (fat-free mass (FFM), body fat mass, and body fatness (percentage fat)) using DXA, BIA, and multifrequency bioelectrical impedance spectroscopy (BIS) [42] showed that FFM, body fat mass, and body fatness were significantly highly correlated between the different methods, with wide limits of agreement among the methods [42]. Additionally, BIA overestimated fat-free mass in lean subjects and underestimated fat mass in overweight subjects [42].

The last method of muscle mass assessment is anthropometry, which is a noninvasive, quantitative, and easily applied method [22]. Anthropometry’s key factors used to determine body composition include height, weight, head circumference, body mass index (BMI), body circumference, and skinfold thickness [22]. In a study of total-body SM mass using DXA, urine creatinine determination, and anthropometric measurements (weight, height, skinfold thickness, and circumference measurements of the mid-upper arm, mid-thigh, and mid-calf) to create an anthropometric prediction model and/or an appropriate equation from creatinine excretion instead of the reference DXA technique to validate SM mass in children and adolescents, the authors concluded that the determination of total-body SM mass in children and adolescents was highly validated with satisfactory confidence via simple anthropometric measurements or 24 h urine creatinine excretion [43].

The most common diagnostic method to detect muscle strength loss is the use of a handheld dynamometer (HHD), which is an efficient, objective, sensitive, and affordable alternative for strength quantitation [44]. When using this method, during a maximum isometric contraction, the examiner holds a tiny portable device against the patient’s limb; the instrument can then assess proximal and distal muscles in all the extremities [45,46]. 

Finally, all the methods used were found to yield accurate results in assessing body composition, especially SM. The differences between the methods depend on patient characteristics (e.g., age and medical diagnosis) and sitting preparation (e.g., availability of equipment and trained staff). The advantages and disadvantages of each method are summarized in Table 1. All the studies that have summarized the findings of diagnostic methods used among hospitalized children are presented in Table 2.

## 4. Feasibility and Accuracy of HGS in Detection MWD among Children

The amount of static force with which the hand can squeeze around a dynamometer can be used to calculate handgrip strength [47]. When standardized techniques and calibrated equipment are used, handgrip strength provides a reliable measurement [48]. Many studies indicate that handgrip strength is a useful method, along with overall nutrition status, to assess MWD by determining muscle strength among healthy children [49,50]. Studies were conducted among healthy children to assess the reliability of a test–retest approach to HGS measurement. The authors showed that HGS is effective for both peak and sustained grip strength and more reliable among preadolescents than among children [49,50].

## 5. Feasibility and Accuracy of HGS in Detection MWD among Hospitalized Children 

As with hospitalized children, there are many studies that use HGS. The HGS z-score for age was assessed in one retrospective cohort using the Pediatric Yorkhill Malnutrition Score (PYMS) to detect muscle function and overall nutritional status. The findings showed that the PYMS was significantly associated with the HGS z-score for age and C-reactive protein, which was inversely related [51]. Another study was cross-sectional and assessed BMI, MUAC, and the HGS z-score for age, comparing them with nutritional screening tools in two groups of healthy and hospitalized children. The authors concluded that HGS did not significantly differ between healthy and hospitalized children. Moreover, a significant association was found between HGS and age and height, whereas a nonsignificant association was found with MUAC [52].

MWD is common among patients with chronic diseases such as cystic fibrosis (CF) [53], Duchenne muscular dystrophy (DMD) [54], chronic kidney disease (CKD) [55], cancer [12], and hematopoietic cell transplantation (HCT) [56]. Muscle assessment was also considered a major part of overall nutritional assessments in many studies that used the HGS z-score as one of the tools to assess muscle strength [23,50,51,52].

A longitudinal pilot study evaluated the associations between the HGS z-score and nutritional status and changes in HGS z-scores post-hospitalization among patients with CF, concluding that no significant relationship exists between the HGS z-score and nutrition status; however, the HGS z-scores during hospitalization were much lower than expected [53]. Compared with children without CF, there was no significant change in HGS among children with CF; however, over time, HGS z-scores were found to be significantly lower [57]. This result means that muscle strength should be assessed separately using a valid, reliable, and feasible tool among children with CF during and after hospitalization [57].

A cross-sectional study evaluating the associations between HGS and functional measures found that HGS may be used in clinical practice as a practical assessment tool to gain immediate insight into the global functional capacity of nonambulatory DMD children [54]. High HGS scores were found among ambulatory children with DMD, whereas low HGS scores were found among nonambulatory patients in a cohort study using a test–retest approach for the reliability of HGS measurements [58]. Thus, HGS is a reliable tool to assess muscle strength among children with DMD during hospitalization.

HGS z-scores are used in children with CKD to assess muscle strength and detect MWD. For example, a cohort study assessing muscle strength found that HGS z-scores in children suffering from CKD with cystinosis were lower than those in children with CKD without cystinosis [55]. Additionally, in another study, according to the degree of CKD, there were differences in the HGS z-score, which found those with stage 2–5 CKD had a lower HGS z-score than those with stage 1 CKD [59]. HGS is a reliable bedside tool for nutritional assessments in children with CKD [55,59,60].

HGS z-scores were used to assess the association between nutritional status and muscle strength in children with cancer. A cross-sectional study showed a strong positive correlation between HGS z-scores, mid-arm muscle circumference (MAMC), and weight as indicators of nutritional status and muscle strength [12], which also showed that low muscle strength was associated with low muscle mass [12].

Furthermore, HGS z-scores were used as a nutritional assessment tool for children with HCT in a cross-sectional study to test the feasibility of incorporating nutrition-focused physical examinations and measures of HGS as part of a comprehensive nutritional assessment. The study found significantly lower HGS z-scores in the undernutrition group than in the normal and overnutrition groups [23,56,61]

A summary of the findings of all the above-mentioned studies is presented in Table 3, which shows that HGS has been frequently used in field research conducted on hospitalized children to assess the overall nutritional status and muscle strength in particular. However, almost all these studies recruited children with a single chronic disease and supported the use of HGS as a clinically accurate and feasible tool in hospital settings among children with certain chronic diseases [12,53,54,55,56]. We cannot generalize these results to all hospitalized children due to a lack of cutoff points to determine MWD, as well as the need to extract data relevant to age, gender, ethnicity, pubertal phases, and clinical population.

## 6. Conclusions

Muscle mass and strength are two criteria that can be used to detect MWD, and several diagnostic methods are used to assess muscle status. The most widely preferred method of analysis is DEX, followed by CT, MRI, BIA, and anthropometry. Recently, HGS has been commonly used as an indicator of muscle strength. HGS is considered one of the best diagnostic methods and is recommended for use with children older than six years, as it is an efficient, objective, quick, easy, and affordable alternative for strength quantitation. Compared with healthy children, HGS is considered an accurate and feasible method to assess muscle strength in hospitalized children, also in children with chronic disease, and it can serve as an indicator of the overall nutritional status. However, longitudinal cohort studies and trials are required to exclusively assess muscle strength among hospitalized children with either chronic or acute disease using HGS in combination with other methods in different intervals upon admission and during hospitalization and to determine cutoffs to detect muscle wasting.

## Figures and Tables

**Figure 1 children-10-00795-f001:**
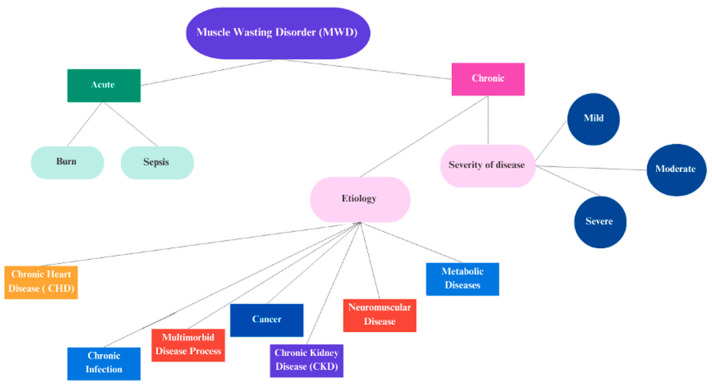
MWD classification is based on etiology and disease severity.

**Figure 2 children-10-00795-f002:**
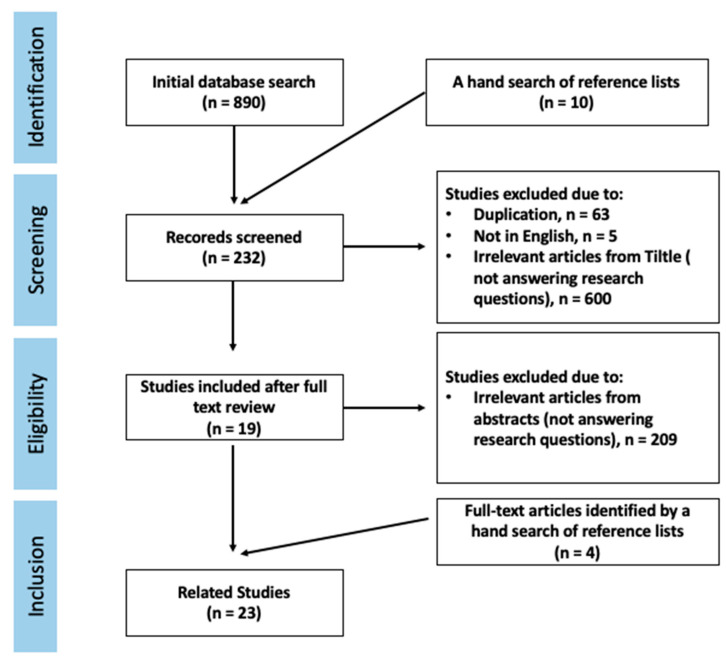
Summary of the search and screening processes used in this narrative review.

**Table 1 children-10-00795-t001:** Differences between diagnostic methods for muscle-wasting disorder.

	Definition	Advantage	Disadvantage
Muscle Mass			
Ultrasonography	A noninvasive and real-time tool for visualizing normal and diseased muscle tissue.	No radiation exposure or sedation, cost-effective, easily applicable, and safe.	Ultrasound equipment required. Restricted for some subjects with fluid–electrolyte imbalance.
Magnetic resonance imaging (MRI)	A noninvasive technique for identifying the muscle-wasting disorder. Provides precise images of internal body structures.	The gold standard, no radiation, precise, and accurate.	Cost-restricted, equipment required, may require sedation.
Computed tomography (CT)	Generates several images or images of the interior of the body including muscle mass	The gold standard, precise, accurate.	Cost-restricted, equipment required, radiation, may require sedation.
Dual-energy X-ray (DXA)	Produces images of the inside of the body, often the hips and lower body. Estimates skeletal muscle.	Simple, quick, no anesthesia required given a total amount of skeletal muscle.	May require sedation, and the radiation dose for this procedure varies; not able to directly measure the amount of skeletal muscle.
Bioelectrical impedance analysis (BIA)	Allows the determination of the fat-free mass and total body water in subjects without significant fluid and electrolyte abnormalities.	Simple, quick, cost-effective, no sedation required, safe.	Restricted for some subjects with fluid–electrolyte imbalance, equipment required, underestimates fat mass, and overestimates fat-free mass.
Anthropometry	Noninvasive method. Elements of anthropometry are height, weight, head circumference, body mass index (BMI), body circumferences to assess for adiposity (waist, hip, and limbs), and skinfold thickness.	Quantitative measurements, simple, and quick.	Restricted for some subjects with fluid–electrolyte imbalance
Muscle Strength			
Handheld dynamometer (HHD)	A small portable device can be used to test both proximal and distal muscles in all extremities.	An efficient, objective, quick, easy, affordable alternative for strength quantitation.	May be difficult when testing large muscles or muscle groups in the lower extremities, sensitive.

**Table 2 children-10-00795-t002:** Studies exploring the diagnostic methods used among hospitalized children.

	Study	Study Design	Population	Sample Size	Age, Year	Control Group and Sample Size	Body Composition/Methodology	Clinical Outcome
	Muscle Mass							
Ultrasonography	[26]	Prospective, observational	Preterm infants	N = 44	Term-equivalent age	Healthy full-term infants, N = 44	Examined the muscle thickness and subcutaneous fat thickness in three different muscles and muscle groups: two proximal (biceps brachii and quadriceps femoris) and one distal (anterior tibial).	Ultrasonography showed lower skeletal muscle mass in preterm versus full-term infants. The muscle thicknesses in the proximal and distal districts were statistically significantly smaller in prematurely born than term-born infants.
	[27]	Case–control	SCP	N = 40	Ages 4–14 years	TD children	Ultrasound scans were taken of the medial gastrocnemius.	Significant differences in skeletal muscle composition were found between children with SCP and their TD peers. Altered tissue composition was observed in children with SCP visualized using ultrasound.
MRI	[18]	Cross-sectional	DMD	N = 26	Ages 6–32 years	N = 12	Spirometry was performed in DMD patients on the same day as MRI. Patients and controls were scanned with a 3-time MRI system.	The results suggest that MRI might represent a new and noninvasive tool for the functional and structural assessment of the diaphragm.
	[33]	Retrospective cohorts	NAFLD	N = 100	Age < 20 years	MRI evidence of hepatic steatosis, N = 236.	Muscle mass was estimated in all patients by measuring MRI-based tPMSA and correcting for height (tPMSA index = tPMSA/height2).	tPMSA index was independently associated with both imaging and histological features of hepatic steatosis severity in children.
CT	[19]	Retrospective	Neuroblastoma	N = 29	Ages 0–18 years	N/A	Cross-sectional areas of skeletal muscle, IMAT, VAT, SAT, and skeletal muscle density at the level of the third lumbar vertebra were examined.	CT scans obtained during standard care provided insight into the direction and timing of changes in skeletal muscle and different types of adipose tissue. Small increases in skeletal muscle (p = 0.029), skeletal muscle density (p = 0.002), and IMAT (p < 0.001) were observed. Rapid increases in VAT (p < 0.001) and SAT (p = 0.001) were observed early during treatment with the highest volumes after six cycles of chemotherapy.
DXA	[20]	Retrospective	Varied in the pubertal maturation stage.	N = 99	Ages 5–17 years		SM estimated using whole-body MRI was used as the reference. The adult SM model was not accurate for subjects below Tanner stage 5 (N = 65; ages 5–14 years). New pediatric SM prediction models were therefore developed and validated in a separate group (N = 18).	The adult DXA skeletal muscle (SM) prediction model was valid in subjects at Tanner stage 5 but significantly (p < 0.001) overestimated SM in subjects below Tanner stage 5. New SM prediction formulas were developed with ALST estimates using DXA as the main predictor variable (e.g., model 1, ALST alone: R-2 = 0.982, SEE = 0.565 kg, p < 0.001).
BIA	[42]	Tool evaluation	Healthy children	N = 61	Ages 8–11 years	N/A	Level of agreement between body composition measurements using DXA, BIA, and multifrequency BIS. FFM, body fat mass, and body fatness (percentage fat) were measured using DXA, BIA, and BIS.	FFM, body fat mass, and body fatness were highly correlated (r = 0.73–0.96, p > 0.0001) between the different methods. Bland–Altman comparison showed wide limits of agreement between the methods BIA overestimated fat-free mass in lean subjects and underestimated fat mass in overweight subjects more than BIS
Anthropometry	[43]	Cross-sectional	Healthy, nonobese subjects	N = 39	Ages 7–16 years	N = 20 adults ages 20–24 years	Total-body SM was assessed using DXA and UCrn determination and anthropometric measurements of weight, height, skinfold thickness, and circumference measurements of the mid-upper arm, mid-thigh, and mid-calf.	Besides the DXA technique, the determination of total-body skeletal muscle mass in children and adolescents can be highly validated with satisfactory confidence using simple anthropometric measurements or 24 h urine creatinine excretion.

SCP: spastic cerebral palsy, TD: typically developing, MRI: magnetic resonance imaging, DMD: Duchenne muscular dystrophy, NAFLD: nonalcoholic fatty liver disease, tPMSA: total psoas muscle surface area, CT: computed tomography, DXA: dual-energy X-ray, IMAT: intermuscular adipose tissue, SAT: subcutaneous adipose tissue, VAT: visceral adipose tissue, SM: skeletal muscle, ALST: appendicular lean soft tissue, BIA: bioelectrical impedance analysis, BIS: bioelectrical impedance spectroscopy, FFM: fat-free mass, UCrn: urine creatinine.

**Table 3 children-10-00795-t003:** Studies that used HGS z-scores to address muscle nutritional status and muscle function among healthy and hospitalized children with chronic disease.

Study	Study Design	Population	Age, Year	Sample Size	Control Group and Sample Size	Body Composition Methodology	Definition of Malnutrition/Muscle Wasting	Clinical Outcome
[49]	Test–retest study	Healthy children	6-, 10-, and 14-year-olds	N = 58	Compared reliability for 3 groups: N = 19 (6 years), N = 20 (10 years), N = 19 (14 years).	HGS	Test–retest reliability	HGS ↑ More reliable in 6- and 14-year-olds than 10-year-olds. ↑ Reliability was good for both peak and sustained grip strength.
[61]	2-year longitudinal cohort study	Students	Age = 9.2 years: 54% female; 83% white)	N = 474	N/A	HGS, BMI, CVD health	HGS, BMI	Low HGS had a significant association with a high prevalence of health decline or poor health persistence. Adolescents who were strong had odds for health maintenance (OR 3.54; 95% CI 1.80–6.97) and health improvement (OR 1.30; 95% CI 1.05–1.60), even after adjustment for baseline fat-free mass index, cardiorespiratory fitness, and objectively measured physical activity.
[50]	Test–retest study	Healthy children	Ages 7–13 years	N = 338	Divided into two groups based on age: children (7–9 years old) and preadolescents (10–13 years old).	HGS	Test–retest reliability	Childhood age influences the difference between the test and retest of the HGS measurement. ↓ HGS measurement using a digital handgrip dynamometer is less reliable among preadolescents than children.
[48]	Evaluation tools study	Healthy	Ages 6–13 years	N = 290	N/A	TE, coefficient of variation, and SWC were calculated.	HGS testing protocol 3 times within a 7-day period.	Changes in HGS were greater than in the TE and SWC, which can be considered real changes of practical significance.
[51]	Retrospective cohort	SC	ages 5–16 years	N = 595 [53]	HC, N = 535 for the development of HGS centile charts	HGS z-scores for age	Using PYMS score	High PYMS scores were significantly associated with low HGS z-scores for age. HGS z-scores were significantly inversely related to plasma CRP.
[52]	Cross-sectional nonequivalent control group design study	Hospitalized patients	Ages 6–14 years	N = 109	Nonhospitalized patients = 110	BMI, MUAC, and HGS z-scores	Nutrition screening tool	HGS did not differ significantly between hospitalized and nonhospitalized. A significant association between HGS and age and height, and a nonsignificant association with MUAC.
[23]	Longitudinal cohort study	Hospitalized patients	≥6 years	N = 89	On admission vs. on discharge	HGS	BMI z-scores	↓ HGS marker of undernutrition
[53]	Longitudinal pilot study	Children with CF	Ages 6–18 years	N = 23	5 months pre-hospitalization, 5–7 days post-hospitalization, and 6 weeks during hospitalization	HGS z-scores and arm anthropometrics	Nutrition screening tool and BMI z-scores	No significant relationship was observed between HGS and nutrition status (BMI z-scores and nutrition risk scores). HGS z-scores at hospitalization were much lower than the standard even though mean BMI z-scores classified participants as having normal nutrition status.
[57]	Cohort study	Medically stable youth with CF	Ages 6–21 years	N = 201	HGS reference tables were created by merging data from the NHANES 2011–2012 and 2013–2014 survey cycles, resulting in HGS measurements from 4672 individuals, ages 6–19 years.	HGS, BMI, body composition (lean body mass and fat-free mass)	HGS and BMI	HGS is reliable, less expensive, and clinically feasible for body composition measurements in monitoring nutrition status.
[62]	Prospective longitudinal study	Children with CF	Ages 6–18 years	N = 75	Non-CF group (N= 76)	MUAC, tricep skinfold, and HGS	Nutrition screening tool	The rate of change in HGS z-scores in both groups was not significant (p = 0.15). HGS z-scores significantly decreased over time in children with CFRD versus children without CFRD.
[56]	Descriptive cross-sectional study	Pediatric HCT	Ages 2–25 years	N = 36		HGS, nutrition-focused physical examination, and online food and activity surveys	BMI, nutrition-focused physical examination, divided into undernutrition, normal, and overnutrition groups	↓ HGS measurements in the undernutrition group was significantly more than those in the normal and overnutrition groups. Comprehensive nutritional assessments and HGS measurements are feasible, noninvasive, easy to perform, and inform both under- and overnutrition in pediatric HCT survivors.
[54]	Cross-sectional study	DMD in children	Ages 5–18 years	N = 38	N/A	HGS, the Turkish version of EK2 for global functional capacity, PUL for upper limb functional performance and the ABILHAND-Kids for hand ability.	Accuracy	HGS was found to be correlated with the EK2 (p < 0.05). HGS may be used in clinical practice as a practical assessment tool to gain immediate insight into the global functional capacity of nonambulatory DMD children.
[58]	Cohort	DMD patients	Ages 5.0–28.7 years	N = 202	92 patients were ambulatory and 110 were nonambulatory	Test–retest reliability of HGS measurements, MyoGrip device	HGS	HGS in ambulatory patients was higher than in nonambulatory patients.
[55]	Cohort	Pt with CKD with cystinosis	All patients > 6 years of age including adults	N = 76	Healthy control subjects, similar CKD stage without cystinosis.	HGS z-score, eGFR	HGS z-score	No significant correlation was observed between eGFR and grip strength z-score. CKD with cystinosis exhibited a mean HGS z-score of −2.1 (SD, 1.1), which is lower ↓ than that found in patients with CKD without cystinosis.
[59]	Prospective cohort	Pt with CKD	1 to 16 years	N = 411	Healthy control subjects	HGS z-score, eGFR	HGS z-score	HGS z-score among CKD patients with CKD stages 2 through 5 was significantly lower than ↓ HGS z-scores for CKD stage 1. Compared with healthy controls, CKD participants had a ↓ Lower HGS z-score
[12]	A cross-sectional study	Cancer patients	Ages 6–19 years	N = 63	The sample was stratified by age group: 6–9 years, 10–14 years, and 15–19 years.	Anthropometric (body weight, height, MUAC, and TSF), BMI, MAMC, and HGS	Anthropometric (body weight, height, MUAC, and TSF), BMI, MAMC, and HGS	A strong positive correlation was observed between HGS and MAMC and weight.

HGS: handgrip strength, BMI: body mass index, CVD: cardiovascular disease, TE: typical error of measurement, SWC: smallest worthwhile change, HC: healthy children, SC: sick children, PYMS: Pediatric Yorkhill Malnutrition Score, CRP: C-reactive protein, MUAC: mid-upper arm circumference, CF: cystic fibrosis, HCT: hematopoietic cell transplantation, DMD: Duchenne muscular dystrophy, EK2: Egen Klassifikation Scale Version 2, PUL: performance of the upper limb, CKD: chronic kidney disease, eGFR: glomerular filtration rate, TSF: tricipital skinfold, MAMC: mid-arm muscle circumference.

## Data Availability

Not applicable.

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
