# Peer review of "Muscle Wasting among Hospitalized Children: A Narrative Review of the Feasibility and Accuracy of Diagnostic Methods"

_children, 2023, doi:10.3390/children10050795_

Round 1
Reviewer 1 Report
The manuscript entitled “Muscle Wasting among Hospitalized Children: Review of Feasibility and Accuracy of Diagnostic Methods” reviewer the most feasible and accurate diagnostic methods for detecting muscle wasting among hospitalized children. The review seems comprehensive, but I think the manuscript did not totally relate to its title.
Here are some suggestions as below
1. The major concern was that the title described the article belongs to a review article, however, in the later part of the manuscript was all about handgrips strength.
The author shall add more information why they separate HGS from other diagnostic methods.
Line 27-28
Why did the author emphasize the important of the skeletal muscle in breathing? In the later part of the manuscript, I did not notice the explanation from the MWD with breathing.
Figure 1
The word is too small to be read. Please consider revising.
The author shall show the PRISMA diagram of the whole review process and results, with a figure or in the supplementary file.
Line 84-85
There are many definitions regarding muscle function and muscle quality. Are both the term muscle mass and muscle strength belonging to muscle function? The author can revisit the definition with adequate references.
Line 93
What is the difference in muscle composition through ultrasonography? By elastography or other parameters? Please be more specific.
Table 1
Disadvantage
“Ultrasound equipment is necessary” it is the same meaning with “equipment required” in the MRI and CT. Please unify all the similar description in the table to make it simple and clear.
Table 2
Why would the author list Study and Country as a column? Is there any important issue regarding the country? Also in the ultrasonography, Florence is not a country.
The author may consider use another way to express.
Some of the enrolled studies has wide range of age, for example, 1ge 6-32 years in the reference 9. I wonder how wound the author define the age range of children?
The author should unify the use of DXA and DEXA in the table. DXA is a preference use.
Line 181
The paragraph could be split into 2 paragraphs for better reading. Also in the paragraph (Line 157-168), the author had discussed handgrip strength. What is the difference between these two paragraphs.
Also, how would the author extend to the following paragraph to particularly use HGS in MWD detection is not clearly mentioned.
The paragraph in section 5 is too long and hard to read. The author should summarize important finding and related clinical implication from individual study.
Section 6
The section can be merged into previous paragraph, since there was not much information in this section.
Conclusion
The conclusion did not match to the title indeed. If the authors want to express that HGS is an important tool, then they should revise the title and explain the purpose of the study.
I think the conclusion is not adequate from the title and the whole manuscript.
Reviewer 2 Report
Interesting Review of Feasibility and Accuracy of Diagnostic Methods of Muscle Wasting. My comments: Please correct the style and punctuation throughout the text. Abstract correct. Introduction Lines 27-32: Please add more articles confirming Muscle Wasting. A broader review of the literature should be done, there are more thematically related articles. The purpose of the work is clearly presented. Materials and Methods : Search methods for related articles should be improved. For now, they are not all developed. More related articles should be presented. Figures and tables are good and legible. Conclusion Correct.Author Response
Please see the attachment

Round 2
Reviewer 1 Report
Thanks for your revision
Author Response
Response to Reviewers
Manuscript ID: Children-2322903
Type of manuscript: Review
Manuscript Title: Muscle Wasting among Hospitalized Children: a Narrative Review of the
Feasibility and Accuracy of Diagnostic Methods
We thank the reviewers for their careful examination of the manuscript and appreciate the useful suggestions to improve the quality of our paper. Our point-by-point response to the reviewer’s comments is given below. Changes in the manuscript are indicated. Please note that the line numbers below refer to the revised manuscript.
Reviewer (1)
I appreciate the opportunity to review the manuscript entitled “Muscle Wasting among Hospitalized Children: Review of Feasibility and Accuracy of Diagnostic Methods Especially Handgrip Strength” submitted to the journal Children. The authors provide an overview of the feasibility and precision of diagnostic methods of MWD, especially handgrip strength as a primary method to detect weakness in muscle strength and delay the progression of MWD, among hospitalized children.
Reviewer Comments:
- This is a narrative review and should be stated in the title. Contrary to
reviewer 1 I would not include the focus on HGS in the title, since I believe
this is a finding of the review, not an a priori strategy. Would suggest
"Muscle Wasting among Hospitalized Children: a Narrative Review of the
Feasibility and Accuracy of Diagnostic Methods".
We appreciate the reviewer’s comment. The title is now modified to the manuscript file.
- The authors have included a search strategy figure in their response to
reviewer 1, but have not included it in the paper or even as supplementary
material in what I have read of the latest version. Even if this is a
narrative review, the credibility and strength of the review ultimately rests
on the strategy. Therefore this figure should be integrated at least as
supplementary material. Also, the search strategy should include clearer
boolean connectors (I'm guessing ORs and ANDs). Finally, I strongly doubt
Pubmed yielded only 14 results (I used a simple combination meaningful
towards to the review aims and identified a couple of hundred papers) and I
would request this be revised.
We thank the reviewer for their comment. The method is now modified to the manuscript file, lines 85-105.
- The authors are correct in saying they do not have to follow PRISMA guidelines or provide a formal flowchart given the narrative review, however the readers must be confident about how the authors limited the risk of bias in exploring relevant literature.
Thank you for the suggestion. We have a formal flowchart to describe the search and screening processes used in this narrative review, lines 106-108.
- Figure 1 is inappropriate for muscle wasting in children, this is a paediatric journal. Esp. COPD and ageing have no place in children. Please provide a figure on MWD in CHILDREN.
We appreciate the reviewer’s comment. Figure 1 is now modified to the manuscript file, lines 59-60.
- Table of for certain difficult to read (esp. crowding of columns in table 2),
please correct. Also title of table 2 is not acceptable "All studies that
have summarized the findings of diagnostic methods used among hospitalized
children." Remove the term "all" since you have not performed a systematic
review. Also these cited papers are not reviews themselves so they do not
"summarize" findings. Find a more appropriate term.
We appreciate the reviewer’s comments. The tables were adjusted, and the title of Table 2 was modified.
We would like to express our appreciation for the referee again for taking the time to review our manuscript.
The revised manuscript is attached.
